# FusionDTI: Fine-grained Binding Discovery with Token-level Fusion for Drug-Target Interaction

## Abstract

Predicting drug-target interaction (DTI) is critical in the drug discovery process. Despite remarkable advances in recent DTI models through the integration of representations from diverse drug and target encoders, such models often struggle to capture the fine-grained interactions between drugs and protein, i.e. the binding of specific drug atoms (or substructures) and key amino acids of proteins, which is crucial for understanding the binding mechanisms and optimising drug design. To address this issue, this paper introduces a novel model, called FusionDTI, which uses a token-level **Fusion** module to effectively learn fine-grained information for **D**rug-**T**arget **I**nteraction. In particular, our FusionDTI model uses the SELFIES representation of drugs to mitigate sequence fragment invalidation and incorporates the structure-aware (SA) vocabulary of target proteins to address the limitation of amino acid sequences in structural information, additionally leveraging pre-trained language models extensively trained on large-scale biomedical datasets as encoders to capture the complex information of drugs and targets. Experiments on three well-known benchmark datasets show that our proposed FusionDTI model achieves the best performance in DTI prediction compared with eight existing state-of-the-art baselines. Furthermore, our case study indicates that FusionDTI could highlight the potential binding sites, enhancing the explainability of the DTI prediction.

## 1 Introduction

The task of predicting drug-target interactions (DTI) plays a pivotal role in the drug discovery progress, as it helps identify potential therapeutic effects of drugs on biological targets facilitating the development of effective treatments (Askr et al., 2023). DTI fundamentally relies on the binding of specific drug atoms (or substructures) and key amino acids of proteins (Schenone et al., 2013). In particular, each binding site is an interaction between a single amino acid and a single drug atom, which we refer to as a fine-grained interaction. For instance, Figure 1 B demonstrates the interaction between *HIV-1 protease* and the drug *lopinavir*. A critical component of this interaction is the formation of a hydrogen bond between a ketone group in lopinavir (represented in the SELFIES (Krenn et al., 2022) notation as [C][=O]) and the side chain of an aspartate residue Asp25 (i.e. Dd) within the protease (Brik & Wong, 2003; Chandwani & Shuter, 2008). Therefore, capturing such fine-grained interaction information during the fusion of drug and target representations is crucial for building effective DTI prediction models (Yazdani-Jahromi et al., 2022; Wu et al., 2022; Peng et al., 2024; Zeng et al., 2024).

To obtain representations of drugs and targets for the DTI task, some previous studies (Lee et al., 2019; Nguyen et al., 2021) have used graph neural networks (GNNs) or convolutional neural networks (CNNs) using a fixed-size window, potentially leading to a loss of contextual information, especially when drugs and targets are in a long-term sequence. These models directly concatenate the representations together to make predictions without considering fine-grained interactions. More recently, some computational models (Huang et al., 2021; Bai et al., 2023) employed the fusion module (e.g. Deep Interactive Inference Network (DIIN) (Gong et al., 2018) and Bilinear Attention Network (BAN) (Kim et al., 2018)) to obtain fine-grained interaction information and the 3-mer approach that binds three amino acids together as a target binding site to address the lack of structural information in the amino acid sequence. While useful for highlighting possible regions of interaction, these models

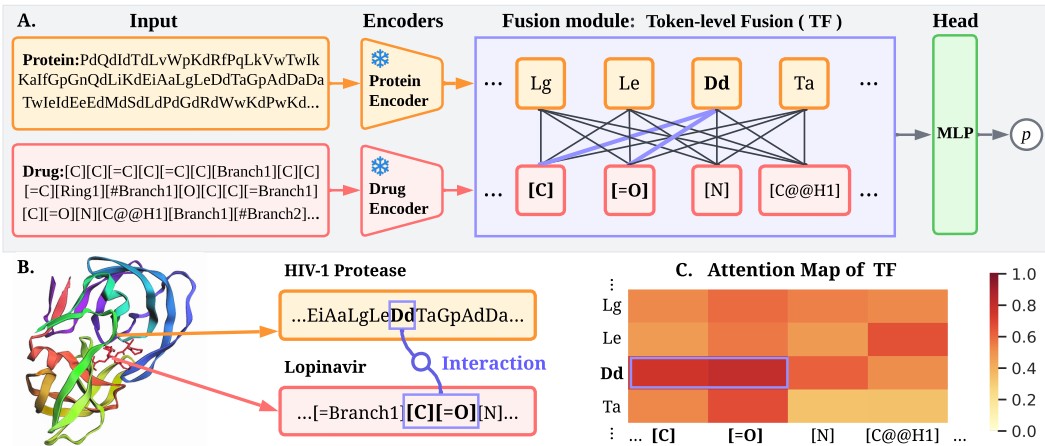

Figure 1: **A**. An illustration of the FusionDTI model contains frozen encoders, the fusion module, and the classifier. The token-level fusion (TF) focuses on fine-grained interactions between tokens within and across sequences. **B**. This is a token-level interaction instance of HIV-1 protease and lopinavir. Lopinavir forms a hydrogen bond with residue Dd (Asp25) in the active site of the protease via its ketone molecule ([C][=O]). **C**. The attention map of TF visualises the weight between tokens, indicating the contribution of each drug atom and residue to the final prediction result.

do not offer the sufficient granularity needed to gauge the specifics of binding sites, as each binding site only contains one residue (Schenone et al., 2013). Therefore, obtaining contextual representations of drugs and targets and capturing fine-grained interaction information for DTI remains challenging.

To address these challenges, we propose a novel model (called FusionDTI) with a Token-level Fusion (TF) module for an effective learning of fine-grained interactions between drugs and targets. In particular, our FusionDTI model utilises two pre-trained language models (PLMs), namely Saport (Su et al., 2023) as the protein encoder that is able to integrate both residue tokens with structure token; and SELFormer (Yüksel et al., 2023) as the drug encoder to ensure that each drug is valid and contains structural information. To effectively learn fine-grained information from these contextual representations of drugs and targets, we explore two strategies for the TF module, i.e. Bilinear Attention Network (BAN) (Kim et al., 2018) and Cross Attention Network (CAN) (Li et al., 2021; Vaswani et al., 2017), to find the best approach for integrating the rich contextual embeddings derived from Saport and SELFormer. We conduct a comprehensive performance comparison against eight existing state-of-the-art DTI prediction models. The results show that our proposed model achieves about 6% accuracy improvement over the best baseline on the BindingDB dataset. The main contributions of our study are as follows:

- We propose FusionDTI, a novel model that leverages PLMs to encode drug SELFIES and protein residue and structure for rich semantic representations and uses the token-level fusion to obtain fine-grained interaction information between drugs and targets effectively.

- We compare two TF modules: CAN and BAN and analyse the influence of fusion scales based on FusionDTI, demonstrating that CAN is superior for DTI prediction both in terms of effectiveness and efficiency.

- We conduct a case study of three drug-target pairs by FusionDTI to evaluate whether potential binding sites would be highlighted for the DTI prediction explainability.

## 2 RELATED WORK

### 2.1 DRUG-TARGET INTERACTION PREDICTION

DTI prediction serves as an important step in the process of drug discovery (Dara et al., 2022). Traditional biomedical measurements from wet experiments are reliable but have a notably high cost and time-consuming development cycle, preventing their application on large-scale data (Zitnik et al.,

2019). In contrast, identifying high-confidence DTI pairs by computational models markedly narrow down the search scope of drug candidate libraries, and aims to identify drugs most likely to bind to a target. Support vector machine (SVM) (Cortes & Vapnik, 1995) and random forest (RF) (Ho, 1995) are two traditional computational models for DTI by concatenating fingerprint ECFP4 (Rogers & Hahn, 2010) and PSC features (Cao et al., 2013). Later works focused on representation learning approaches, such as CNNs and GNNs Lee et al. (2019); Nguyen et al. (2021). For example, DeepConv-DTI (Lee et al., 2019) employed CNNs and a global max-pooling layer to extract local protein sequence patterns. Similarly, GraphDTA (Nguyen et al., 2021) used GNNs for drug graph encoding and CNNs for protein sequence encoding. Furthermore, MolTrans (Huang et al., 2021) introduced an adaptation of the transformer for encoding, further enhanced by a DIIN module (Gong et al., 2018) to learn fine-grained interactions. Additionally, DrugBAN (Bai et al., 2023) incorporated a deep BAN (Kim et al., 2018) framework with domain adaptation to facilitate explicit pairwise fine-grained interaction learning between drugs and targets. Moreover, BioT5 (Pei et al., 2023) has been proposed as a comprehensive pre-training framework that integrates cross-modelling in biology in the DTI task. More recently, SiamDTI (Zhang et al., 2024) utilises a double-channel network structure to acquire local and global protein information for cross-field DTI prediction. Despite these advances, these models have not proposed an effective way to capture fine-grained interaction information in the DTI.

## 2.2 Drug and Protein Representation

For drug molecules, most existing methods represent the input by the Simplified Molecular Input Line Entry System (SMILES) (Weininger, 1988; Weininger et al., 1989). However, SMILES suffers from numerous problems in terms of validity and robustness, and some valuable information about the drug structure may be lost which may prevent the model from efficiently mining the knowledge hidden in the data reducing the predictive performance of the model (Krenn et al., 2022). In particular, SMILES fragments are often invalid and inconsistent with the substructural information of the drug. To address the limitations of SMILES, we apply SELFIES (Krenn et al., 2022), a string-based representation that circumvents the issue of robustness and that always generates valid molecular graphs for each character (Krenn et al., 2022).

Regarding proteins, the conventional approach uses amino acid sequences as model inputs (Huang et al., 2021; Bai et al., 2023), overlooking the crucial structural information of the protein. Inspired by the SA vocabulary of SaProt (Su et al., 2023), the SaProt enhances inputs by amalgamating each residue from the amino acid sequence with a 3D geometric feature that is obtained by encoding the structure information of the protein using Foldseek (Van Kempen et al., 2024). This innovative combination offers richer protein representations through the SA vocabulary, contributing to the discovery of fine-grained interactions. Our proposed model employs SELFIES for drug encoding and uses SaProt encoding for proteins to generate the semantic representations for both drugs and targets.

## 2.3 Molecular and Protein Language Models

Molecular language models that train on the large-scale molecular corpus to capture the subtleties of chemical structures and their biological activities have set new standards in encoding chemical compounds achieving meaningful representations Ying et al. (2021); Rong et al. (2020). For example, ChemBERTa-2 (Ahmad et al., 2022) used RoBERTa-based architectures to capture intricate molecular patterns, significantly enhancing the precision of property prediction. Subsequently, MoLFormer (Ross et al., 2022) focused on leveraging the self-attention mechanism to interpret the complex, non-linear interactions within molecules, while SELFormer (Yüksel et al., 2023) employed SELFIES, ensuring valid and interpretable chemical structures.

Protein language models have revolutionized the way we understand and represent protein sequences, offering richer semantic representations (Elnaggar et al., 2021; Lin et al., 2023; Su et al., 2023). These models leverage the vast corpus of biological sequence data, learning intricate patterns and features that define the protein functionality and interactions. ProtBERT (Elnaggar et al., 2021) and ESM (Lin et al., 2023) applied a transformer architecture to protein sequences, capturing the complex relationships between amino acids. Saport (Su et al., 2023) further enhanced this approach by integrating SA vocabularies to provide protein structure information. Furthermore, SaprotHub (Su

et al., 2024) offers a platform that enables biologists to train, deploy, and share protein models efficiently. Importantly, our FusionDTI is flexible enough to use each of them as a protein encoder.

## 3 METHODOLOGY

### 3.1 MODEL ARCHITECTURE

Given a sequence-based input drug-target pair, the DTI prediction task aims to predict an interaction probability score $p \in [0, 1]$ between the given drug-target pair, which is typically achieved through learning a joint representation $\mathbf{F}$ space from the given sequence-based inputs. To address the DTI task and effectively capture fine-grained interaction, we proposed a novel model, called FusionDTI, which is a bi-encoder model Liu et al. (2021) with a fusion module that fuses the representations of drugs and targets. The overall framework of FusionDTI is illustrated in Figure 1 A. In general, FusionDTI takes sequence-based inputs of drugs and targets, which are encoded into token-level representation vectors by two frozen encoders. Then, a fusion module fuses the representations to capture fine-grained binding information for a final prediction through a prediction head.

**Input**: The initial inputs of drugs and targets are string-based representations. For protein $\mathcal{P}$, the SA vocabulary (Su et al., 2023; Van Kempen et al., 2024) is employed, where each residue is replaced by one of 441 SA vocabularies that bind an amino acid to a 3D geometric feature to address the lack of structural information in amino acid sequences. For drug $\mathcal{D}$, as mentioned in the previous section, we use the SELFIES, which is a formal syntax that always generates valid molecular graphs (Krenn et al., 2022). We provide the steps and code for obtaining SA and SELFIES sequences in Appendix A.3.

**Encoder**: The proposed model contains two frozen encoders: Saport (Su et al., 2023) and SELFormer (Yüksel et al., 2023), which generate a drug representation $\mathbf{D}$ and a protein representation $\mathbf{P}$ separately. It is of note that FusionDTI is flexible enough to easily replace encoders with other advanced PLMs. Furthermore, $\mathbf{D}$ and $\mathbf{P}$ are stored in memory for later-stage online training.

**Fusion module**: In developing FusionDTI, we have investigated two options for the fusion module: BAN and CAN to fuse representations, as indicated in Figure 2. The CAN is utilised to fuse each pair as $\mathbf{D}^*$ and $\mathbf{P}^*$, and then concatenate them into one $\mathbf{F}$ for fine-grained binding information. For BAN, we need to obtain the bilinear attention map and then generate $\mathbf{F}$ through the bilinear pooling layer.

**Prediction head**: Finally, we obtain the probability score $p$ of the DTI prediction by a multilayer perceptron (MLP) classifier trained with the binary cross-entropy loss, i.e. $p = \text{MLP}(\mathbf{F})$.

Since the encoders and the fusion module constitute the key components of our FusionDTI model, we will describe them in detail in the following subsections.

### 3.2 DRUG AND PROTEIN ENCODERS

Employing sequences with detailed biological functions and structures is a critical step in exploring the fine-grained binding of drugs and targets. For drugs, SMILES is the most commonly used input sequence but suffers from invalid sequence segments and potential loss of structural information (Krenn et al., 2022). To address the limitations, we transform SMILES into SELFIES, a formal grammar that generates a valid molecular graph for each element (Krenn et al., 2022). Besides, to address the lack of structural information in the amino acid sequences, we utilise the SA sequence of targets to combine each amino acid with an SA vocabulary by Foldseek (Van Kempen et al., 2024).

PLMs have shown promising achievements in the biomedical domain leveraging transformers since they pay attention to contextual information and are pre-trained on large-scale biomedical databases. Therefore, we utilise Saport (Su et al., 2023) as a protein encoder to encode protein input $\mathcal{P}$ of both the SA sequence and amino acid sequence. Meanwhile, SELFormer (Yüksel et al., 2023) is used as our drug encoder to encode the drug SELFIES input $\mathcal{D}$. Then these encoded protein representation $\mathbf{P}$ and drug representation $\mathbf{D}$ are further used as inputs for the later fusion module (Subsection 3.3). These rich contextual representations ensure that we can explore the fine-grained binding information effectively. To further justify this, we also compare our encoders with other existing protein language models (such as ESM-2 (Lin et al., 2023)) and molecular language models (such as MoLFormer (Ross et al., 2022) and ChemBERTa-2 (Ahmad et al., 2022)), and the results can be found in Section 4.7.

## 3.3 FUSION MODULE

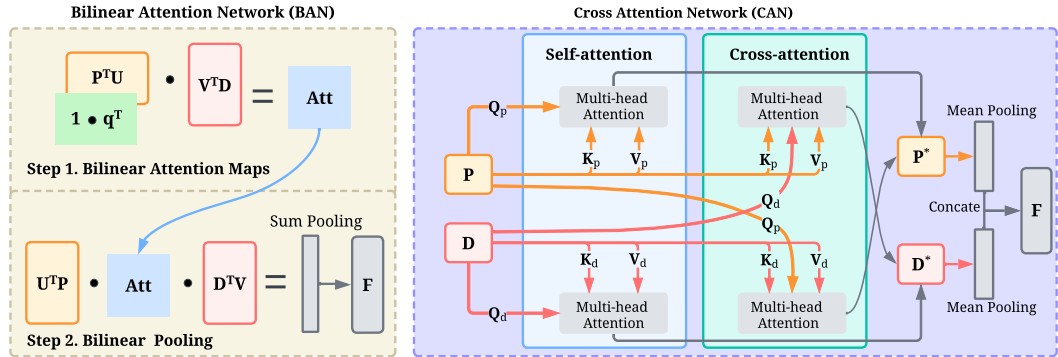

Figure 2: **BAN:** In step 1, the bilinear attention map matrix is obtained by a bilinear interaction modelling via transformation matrices. In step 2, the joint representation $\mathbf{F}$ is generated using the attention map by bilinear pooling via the shared transformation matrices $\mathbf{U}$ and $\mathbf{V}$. **CAN:** It fuses protein and drug representations through multi-head, self-attention and cross-attention. Then fused representations $\mathbf{P}^*$ and $\mathbf{D}^*$ are concatenated into $\mathbf{F}$ after mean pooling.

In order to capture the fine-grained binding information between a drug and a target, our FusionDTI model applies a fusion module to learn token-level interactions between the token representations of drugs and targets encoded by their respective encoders. As shown in Figure 2, two fusion modules inspired by the recent literature (Bai et al., 2023; Xu et al., 2023) are investigated to fuse representations: the Bilinear Attention Network (Kim et al., 2018) and the Cross Attention Network (Li et al., 2021; Vaswani et al., 2017).

### 3.3.1 BILINEAR ATTENTION NETWORK (BAN)

Motivated by DrugBAN (Bai et al., 2023), our model considers BAN (Kim et al., 2018) as an option of the fusion module to learn pairwise fine-grained interactions between drug $\mathbf{D} \in \mathbb{R}^{M \times \phi}$ and target $\mathbf{P} \in \mathbb{R}^{N \times \rho}$, denoted as FusionDTI-BAN. For BAN as indicated in Figure 2, bilinear attention maps are obtained by a bilinear interaction modelling to capture pairwise weights in step 1, and then the bilinear pooling layer to extract a joint representation $\mathbf{F}$. The equation for BAN is shown below:

$$
\begin{aligned}
\mathbf{F} &= \mathrm{BAN}(\mathbf{P}, \mathbf{D}; Att) \\
&= \mathrm{SumPool}(\sigma(\mathbf{P}^\top \mathbf{U}) \cdot Att \cdot \sigma(\mathbf{D}^\top \mathbf{V}), s),
\end{aligned}
\tag{1}
$$

where $\mathbf{U} \in \mathbb{R}^{N \times K}$ and $\mathbf{V} \in \mathbb{R}^{M \times K}$ are transformation matrices for representations. $\mathrm{SumPool}$ is an operation that performs a one-dimensional and non-overlapped sum pooling operation with stride $s$ and $\sigma(\cdot)$ denotes a non-linear activation function with $\mathrm{ReLU}(\cdot)$. $Att \in \mathbb{R}^{\rho \times \phi}$ represents the bilinear attention maps using the Hadamard product and matrix-matrix multiplication and is defined as:

$$
Att = ((\mathbf{1} \cdot \mathbf{q}^\top) \circ \sigma(\mathbf{P}^\top \mathbf{U})) \cdot \sigma(\mathbf{V}^\top \mathbf{D}),
\tag{2}
$$

Here, $\mathbf{1} \in \mathbb{R}^\rho$ is a fixed all-ones vector, $\mathbf{q} \in \mathbb{R}^K$ is a learnable weight vector and $\circ$ denotes the Hadamard product. In this way, pairwise interactions contribute sub-structural pairs to the prediction.

BAN captures the token-level interactions between the protein and drug representations without considering the relationships within each sequence itself, which may limit its ability to understand deeper contextual dependencies.

### 3.3.2 CROSS ATTENTION NETWORK (CAN)

Inspired by ProST (Xu et al., 2023), we also consider CAN as our fusion module to learn fine-grained interaction information of drugs and targets. We denote our FusionDTI model that uses a CAN fusion module as FusionDTI-CAN. By processing $\mathbf{D} \in \mathbb{R}^{m \times h}$ and $\mathbf{P} \in \mathbb{R}^{n \times h}$ separately, the fused drug $\mathbf{D}^* \in \mathbb{R}^{m \times h}$ and target $\mathbf{P}^* \in \mathbb{R}^{n \times h}$ representations are obtained. To synthesise the fine-grained joint

representation $\mathbf{F}$, we employ a pooling aggregation strategy for both $\mathbf{D}^*$ and $\mathbf{P}^*$ independently and then concatenate them as shown in Figure 2. The process is delineated by the following equation:

$$\mathbf{F} = \text{Concat}((\text{MeanPool}(\mathbf{D}^*, \dim = 1), \text{MeanPool}(\mathbf{P}^*, \dim = 1)), 1), \tag{3}$$

where $\text{MeanPool}$ calculates the element-wise mean of all tokens across the sequence dimension, and $\text{Concat}$ denotes the concatenation of the resulting mean vectors. In this context, the multi-head, self-attention and cross-attention mechanisms are used to refine the representations of each residue and atom as below:

$$\mathbf{D}^* = \frac{1}{2} \left[ MHA(\mathbf{Q}_d, \mathbf{K}_d, \mathbf{V}_d) + MHA(\mathbf{Q}_p, \mathbf{K}_d, \mathbf{V}_d) \right], \tag{4}$$

$$\mathbf{P}^* = \frac{1}{2} \left[ MHA(\mathbf{Q}_p, \mathbf{K}_p, \mathbf{V}_p) + MHA(\mathbf{Q}_d, \mathbf{K}_p, \mathbf{V}_p) \right], \tag{5}$$

where $\mathbf{Q}_d, \mathbf{K}_d, \mathbf{V}_d \in \mathbb{R}^{m \times h}$ and $\mathbf{Q}_p, \mathbf{K}_p, \mathbf{V}_p \in \mathbb{R}^{n \times h}$ are the queries, keys and values for drug and target protein, respectively. And $MHA$ denotes the Multi-head Attention mechanism. To guide this process, two distinct sets of projection matrices guide the attention mechanism as follows:

$$\mathbf{Q}_d = \mathbf{D}\mathbf{W}_q^d, \quad \mathbf{K}_d = \mathbf{D}\mathbf{W}_k^d, \quad \mathbf{V}_d = \mathbf{D}\mathbf{W}_v^d, \tag{6}$$

$$\mathbf{Q}_p = \mathbf{P}\mathbf{W}_q^p, \quad \mathbf{K}_p = \mathbf{P}\mathbf{W}_k^p, \quad \mathbf{V}_p = \mathbf{P}\mathbf{W}_v^p, \tag{7}$$

Here, the projection matrices $\mathbf{W}_q^d, \mathbf{W}_k^d, \mathbf{W}_v^d \in \mathbb{R}^{h \times h}$ and $\mathbf{W}_q^p, \mathbf{W}_k^p, \mathbf{W}_v^p \in \mathbb{R}^{h \times h}$ are used to derive the queries, keys and values, respectively.

In summary, our CAN module combines multi-head, self-attention and cross-attention mechanisms to capture dependencies within individual sequences and between different sequences for a more nuanced understanding of interactions. In the results of Sections 4.3 and 4.6, we analyse and compare these two fusion strategies and different fusion scales in detail.

## 4 EXPERIMENTAL SETUP AND RESULTS

### 4.1 DATASETS AND BASELINES

Three public DTI datasets, namely BindingDB (Gilson et al., 2016), BioSNAP (Zitnik et al., 2018) and Human (Liu et al., 2015; Chen et al., 2020), are used for evaluation, where each dataset is split into training, validation, and test sets with a 7:1:2 ratio using two different splitting strategies: in-domain and cross-domain. For the in-domain split, the datasets are randomly divided. For the cross-domain setting, the datasets are split such that the drugs and targets in the test set do not overlap with those in the training set, making it a more challenging scenario where models must generalise to novel drug-target interactions. Since DTI is a binary classification task, we use AUROC (area under the receiver operating characteristic curve) (Bai et al., 2023; Huang et al., 2021) and AUPRC (area under the precision-call curve) (Lee et al., 2019; Nguyen et al., 2021) as the major metrics to evaluate a model's performance.

We compare FusionDTI with eight baseline models in the DTI prediction task. These models include two traditional machine learning methods such as SVM (Cortes & Vapnik, 1995) and Random Forest (RF) (Ho, 1995), as well as five deep learning methods including DeepConv-DTI (Lee et al., 2019), GraphDTA (Nguyen et al., 2021), MolTrans (Huang et al., 2021), DrugBAN (Bai et al., 2023) and SiamDTI (Zhang et al., 2024). The latter five models employ the same two-stage process whereby the drug and target features are initially extracted by specialised encoders before being integrated for prediction. In addition, we also include the BioT5 (Pei et al., 2023) model, which is a biomedical pre-trained language model that could directly predict the DTI. Further details regarding the datasets, baseline models, and the methodology for generating drug SELFIES and protein SA sequences are provided in Appendix A.3.

### 4.2 EFFECTIVENESS EVALUATION FOR DTI PREDICTION

We start by comparing our FusionDTI model (FusionDTI-CAN and FusionDTI-BAN) with eight existing state-of-the-art baselines for DTI prediction on three widely used datasets. Table 1 reports

Table 1: In-domain performance comparison of FusionDTI and the baselines on the BindingDB, Human and BioSNAP datasets (**Best**, Second Best).

| Method | BindingDB | | | Human | | | BioSNAP | | |
|---|---|---|---|---|---|---|---|---|---|
| | AUROC | AUPRC | Accuracy | AUROC | AUPRC | AUROC | AUPRC | Accuracy | |
| SVM | .939±.001 | .928±.002 | .825±.004 | .940±.006 | .920±.009 | .862±.007 | .864±.004 | .777±.011 | |
| RF | .942±.011 | .921±.016 | .880±.012 | .952±.011 | .953±.010 | .860±.005 | .886±.005 | .804±.005 | |
| DeepConv-DTI | .945±.002 | .925±.005 | .882±.007 | .980±.002 | .981±.002 | .886±.006 | .890±.006 | .805±.009 | |
| GraphDTA | .951±.002 | .934±.002 | .888±.005 | .981±.001 | .982±.002 | .887±.008 | .890±.007 | .800±.007 | |
| MolTrans | .952±.002 | .936±.001 | .887±.006 | .980±.002 | .978±.003 | .895±.004 | .897±.005 | .825±.010 | |
| DrugBAN | .960±.001 | .948±.002 | .904±.004 | .982±.002 | .980±.003 | .903±.005 | .902±.004 | .834±.008 | |
| SiamDTI | .961±.002 | .945±.002 | .890±.006 | .970±.002 | .969±.003 | .912±.005 | .910±.003 | .855±.004 | |
| BioT5 | .963±.001 | .952±.001 | .907±.003 | .989±.001 | .985±.002 | .937±.001 | .937±.004 | .874±.001 | |
| FusionDTI-BAN | .975±.002 | .976±.002 | .933±.003 | .984±.002 | .984±.003 | .923±.002 | .921±.002 | .856±.001 | |
| FusionDTI-CAN | **.989±.002** | **.990±.002** | **.961±.002** | **.991±.002** | **.989±.002** | **.951±.002** | **.951±.002** | **.889±.002** | |

Table 2: Cross-domain performance comparison of FusionDTI and the baselines on the BindingDB, Human and BioSNAP datasets (**Best**, Second Best).

| Method | BindingDB | | | Human | | | BioSNAP | | |
|---|---|---|---|---|---|---|---|---|---|
| | AUROC | AUPRC | Accuracy | AUROC | AUPRC | AUROC | AUPRC | Accuracy | |
| SVM | .490±.015 | .460±.001 | .531±.009 | .621±.036 | .637±.009 | .602±.005 | .528±.005 | .513±.011 | |
| RF | .493±.021 | .468±.023 | .535±.012 | .642±.011 | .663±.050 | .590±.015 | .568±.018 | .499±.004 | |
| GraphDTA | .536±.015 | .496±.029 | .472±.009 | .822±.009 | .759±.006 | .618±.005 | .618±.008 | .535±.024 | |
| DeepConv-DTI | .527±.038 | .499±.035 | .490±.027 | .761±.016 | .628±.022 | .645±.022 | .642±.032 | .558±.025 | |
| MolTrans | .554±.024 | .511±.025 | .470±.004 | .810±.021 | .745±.034 | .621±.015 | .608±.022 | .546±.032 | |
| DrugBAN | .604±.027 | .570±.047 | .509±.021 | .833±.020 | .760±.031 | .685±.044 | .713±.041 | .565±.056 | |
| SiamDTI | .627±.027 | .571±.024 | .563±.033 | **.863±.019** | .807±.040 | .718±.055 | .725±.054 | .623±.070 | |
| BioT5 | .651±.002 | .653±.003 | .621±.005 | .856±.003 | **.853±.003** | .720±.008 | .718±.004 | .715±.009 | |
| FusionDTI-BAN | .659±.002 | .663±.002 | .633±.003 | .784±.002 | .790±.003 | .723±.002 | .721±.002 | .756±.001 | |
| FusionDTI-CAN | **.675±.005** | **.676±.012** | **.649±.005** | .801±.037 | .803±.032 | **.748±.021** | **.766±.017** | **.734±.012** | |

the in-domain comparative results. In general, our FusionDTI-CAN model performs the best on all metrics across all three datasets. A key highlight from these results is the exceptional performance of FusionDTI-CAN on the BindingDB dataset, where FusionDTI-CAN demonstrates superior metrics across the board: an AUROC of 0.989, an AUPRC of 0.990, and an accuracy of 96.1%. Note that the main difference between the FusionDTI-CAN model and others is the fusion strategy. Furthermore, despite FusionDTI-BAN and DrugBAN both utilising the same BAN module, FusionDTI-BAN consistently outperforms DrugBAN on all datasets.

However, in-domain classification using random splits holds limited practical significance. Thus, we also evaluate the more challenging cross-domain DTI prediction, where the training data and the test data contain distinct drugs and targets. This setting precludes the use of known drug or target features when making predictions on the test data. As shown in Table 2, the performance of all models is diminished compared to the in-domain setting due to the reduced availability of information. Nevertheless, the FusionDTI-CAN model demonstrates outstanding performance in cross-domain DTI prediction on the BindingDB and BioSNAP datasets, highlighting its robustness in predicting novel drug-target interactions. For instance, on the BindingDB dataset, FusionDTI-CAN achieves the highest metrics with an AUROC of 0.675 and an AUPRC of 0.676. This underscores the effectiveness of the model's fusion strategy in diverse and challenging scenarios. Similarly, despite sharing the BAN module, FusionDTI-BAN continues to outperform DrugBAN, further confirming the effectiveness of the FusionDTI framework in addressing cross-domain prediction challenges.

These findings highlight not only the substantial improvements of FusionDTI over existing approaches but also its effectiveness in capturing fine-grained information on DTI. The key to this success lies in FusionDTI's token-level fusion module, which enables the model to consider fine-grained interactions for each drug-target pair. This fine-grained interaction information aligns closely with biomedical pathways, where binding events often depend on the specific atoms or substructures involved in interactions with residues. Therefore, the model's ability to capture such fine-grained interactions significantly enhances its predictive performance for DTI.

## 4.3 COMPARISON OF THE BAN AND CAN FUSION MODULES

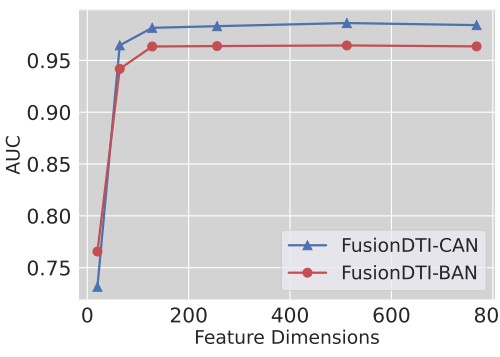

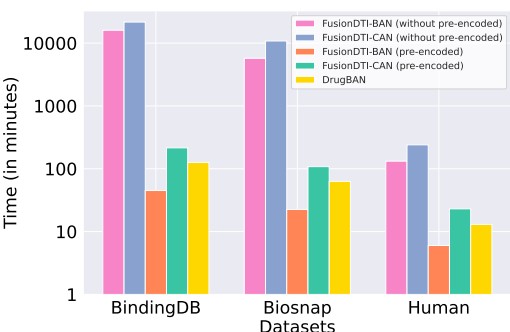

Figure 3: Performance comparison of two fusion strategies: BAN and CAN on BindingDB.

Figure 4: Time comparison on the BindingDB, Human and BioSNAP datasets.

There are two fusion strategies available: BAN and CAN, thus determining which one works better is a key step for establishing FusionDTI's prediction effectiveness. We perform a fair comparison involving the same encoders, classifier and dataset. As shown in Figure 3, we compare BAN and CAN by employing two linear layers to adjust the feature dimensions of the drug and target representations. With the feature dimension increasing, the performance of FusionDTI-CAN continues to rise, while that of FusionDTI-BAN reaches a plateau. When the feature dimension is 512, both of the variants attain their peak positions with an AUC of 0.989 and 0.967, respectively. These results indicate that the CAN module seems to be better suited to the DTI prediction tasks and in capturing fine-grained interaction information. In contrast, BAN may not be able to fully capture fine-grained binding information between proteins and drugs, such as the specific interactions between the drug atoms and residues. Therefore, these findings suggest that the CAN strategy is more effective and adaptable to the complexities involved in DTI prediction, providing superior performance, especially as the feature dimension scales.

## 4.4 EFFICIENCY ANALYSIS

Efficiency in computational models is crucial, particularly when handling large-scale and extensive datasets in drug discovery. Our proposed model stores drug representations and target representations in memory for later online training. As evidenced by Figure 4, FusionDTI-CAN and FusionDTI-BAN with pre-encoded representations process the BindingDB dataset much faster than the non-pre-coded models, approximately 45 minutes and 220 minutes, respectively. This stark difference highlights the advantage of pre-encoded, which eliminates the need for real-time data processing and accelerates the overall throughput. While FusionDTI-BAN and DrugBAN have the same fusion module, the pre-encoded FusionDTI-BAN runs faster and predicts more accurately, as shown in Table 1. In addition, FusionDTI-BAN runs faster than FusionDTI-CAN, indicating that the BAN fusion module is more efficient. Ultimately, FusionDTI-BAN with pre-encoded data stands out as a highly efficient approach, offering substantial benefits in scenarios where exists large-scale data. We further analyse the time complexity in Appendix A.6.

## 4.5 ABLATION STUDY

Table 3: Ablation study of FusionDTI on the BindingDB dataset.

| CAN | AUC | AUPRC | Accuracy |
| --- | --- | --- | --- |
| × | 0.954 | 0.963 | 0.894 |
| ✓ | 0.989 | 0.990 | 0.961 |

Table 4: Comparison of aggregation strategies for CAN on the BindingDB dataset.

| Aggregation | AUC | AUPRC | Accuracy |
| --- | --- | --- | --- |
| CLS | 0.982 | 0.983 | 0.956 |
| Pooling | 0.989 | 0.990 | 0.961 |

The fine-grained interaction of drug and target representations is critical in DTI as it directly impacts the model's ability to infer potential binding sites. For FusionDTI, this interaction is facilitated by

the CAN module, which markedly enhances the predictive accuracy by capturing the fine-grained interaction information between the drugs and targets. Table 3 demonstrates the impact of the CAN module on the prediction performance using the BindingDB dataset. When the fusion module is omitted, the model achieves an AUC of 0.954 and an accuracy of 0.894. Conversely, using the CAN module, there is a significant improvement, with the AUC increasing to 0.989 and the accuracy reaching 0.961. This highlights the effectiveness of the CAN module in improving the inference ability of FusionDTI. Additionally, in Table 4, we compare the performance of two aggregation strategies within the CAN module. The pooling strategy outperforms the CLS-based aggregation, achieving an AUC and AUPRC of 0.989 and 0.990, respectively. This comparison highlights the superior effectiveness of the pooling in aggregating contextual information. Thus, the integration of a CAN module, particularly employing a pooling aggregation strategy, is shown to be essential for making confident and accurate predictions.

## 4.6 ANALYSIS OF FUSION SCALES

In assessing fusion representations, it is critical to determine whether more fine-grained modelling enhances the predictive performance. Thus, we define a grouping function with the parameter **g** (Group size) for averaging tokens within each group before the CAN fusion module. The parameter **g**, representing the number of tokens per group, controls the granularity of the attention mechanism. Specifically, when **g** is set to 1, the fusion operates at the token level, where each token is considered independently. In contrast, when **g** is set to 512, the fusion occurs at a global level, considering the entire embedding as a single unit. We have the flexibility to control the fusion scale for the drug and protein representations, but the token length must be divisible by the group size. As shown in Figure 5, as the number of tokens per group increases from 1 to 512 (Maximum Token Length), the performance of the FusionDTI model declines accordingly. This also aligns with the biomedical rules governing drug-protein interactions, where the principal factor influencing the binding is the interplay between the key atoms or substructures in the drug and primary residues in the protein. Furthermore, the CAN module outperforms BAN consistently at various scale settings, indicating that CAN better accesses the information between the drug and target. Consequently, this supports that the more detailed the interaction information obtained between the drugs and targets by the fusion module, the more beneficial it is for the enhancement of the model's prediction performance.

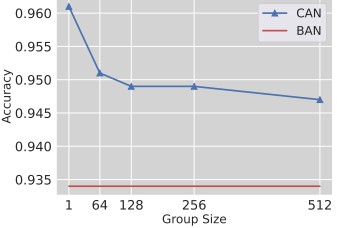

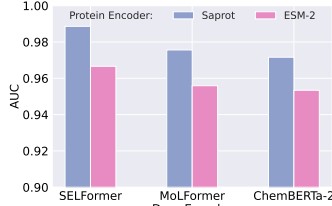

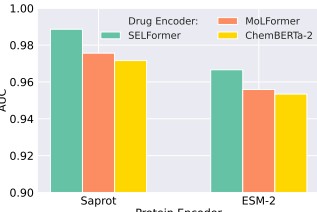

Figure 5: Performance evaluation of fusion scales.

Figure 6: Performance comparison of protein encoders.

Figure 7: Performance comparison of drug encoders.

## 4.7 EVALUATION OF PLMs ENCODING

The protein encoder and drug encoder are fundamental for the token-level fusion of representations, as these encoders are responsible for generating fine-grained representations to better explore interaction information. Our proposed model employs two PLMs encoding two biomedical entities: the drug and protein, respectively. In terms of the protein encoders, Figure 6 compares the the performance of the two protein encoders (SaProt (Su et al., 2023) and ESM-2 (Lin et al., 2023)) in combination with three different drug encoders: ChemBERTa-2 (Ahmad et al., 2022), SELFormer (Yüksel et al., 2023) and MoLFormer (Ross et al., 2022). From the figure, we find that SaProt consistently outperforms ESM-2 when combined with all three drug encoders. As can be seen in Figure 7, SELFormer achieves the best performance in encoding the drug sequences among the three advanced drug encoders. Notably, the top-performing combination is SaProt and SELFormer, hence our proposed FusionDTI uses them as drug and protein encoders.

### 4.8 CASE STUDY

Table 5: FusionDTI predictions: **Bold** represents new predictions versus DrugBAN.

| Drug-Target Interactions |
| --- |
| **EZL - 6QL2:** |
| **1**. sulfonamide oxygen - Leu198, Thr199 and Trp209; |
| **2**. amino group - His94, His96, **His119** and Thr199; |
| **3**. benzothiazole ring - Leu198, Thr200, **Tyr131**, Pro201 and **gln92**; |
| **4**. ethoxy group - **Gln135**; |
| **9YA - 5W8L:** |
| **1**. amino group of sulfonamide - Asp140, Glu191; |
| **2**. sulfonamide oxygen - Asp140, Ile141 and **Val139**; |
| **3**. carboxylic acid oxygens - Arg168, His192, **Asp194** and Thr247; |
| **4**. biphenyl rings - Arg105, Asn137 and **Pro138**; |
| **5**. hydrophobic contact - **Ala237**, **Try238** and **Leu322**; |
| **EJ4 - 4N6H:** |
| **1**. basic nitrogen of ligand - Asp128; |
| **2**. hydrophobic pocket - Tyr308, Ile304 and **Tyr129**; |
| **3**. water molecules - **Tyr129**, Met132, **Trp274**, Try308 and Lys214; |

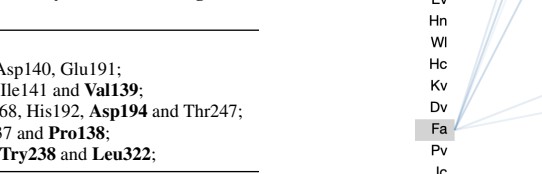

Figure 8: EZL - 6QL2: Fine-grained interactions via attention visualization.

A further strength of FusionDTI to enable explainability, which is critical for drug design efforts, is the visualisation of each token's contribution to the final prediction through cross-attention maps. To compare with the DrugBAN model, we examine three identical pairs of DTI from the Protein Data Bank (PDB) (Berman et al., 2007): (EZL - 6QL2 (Kazokaitė et al., 2019), 9YA - 5W8L (Rai et al., 2017) and EJ4 - 4N6H (Fenalti et al., 2014)). As shown in Table 5, our proposed model predicts additional binding sites (in bold) evidenced by PDB (Berman et al., 2007) in comparison to the DrugBAN model. For instance, to predict the interaction of the drug EZL with the target 6QL2, our proposed model using BertViz (Vig, 2019) highlights potential binding sites as illustrated in Figure 8. Specifically, our CAN module is effective in capturing fine-grained binding information at the token level, as we have successfully predicted the novel binding between Gln92 and the benzothiazole ring (Di Fiore et al., 2008). In particular, we address the lack of structural information on protein sequences by employing the SA vocabulary, which matches each residue to a corresponding 3D feature via Foldseek (Van Kempen et al., 2024). This study highlights the effectiveness of FusionDTI in enhancing performance on the DTI task, thereby supporting more targeted and efficient drug development efforts. In Appendix A.5, we present three pairs of prediction visualisations.

## 5 CONCLUSIONS

With the rapid increase of new diseases and the urgent need for innovative drugs, it is critical to capture and gauge fine-grained interactions, since the binding of specific drug atoms to the main amino acids is key to the DTI task. Despite some achievements, fine-grained interaction information is not effectively captured. To address this challenge, we introduce FusionDTI uses token-level fusion to effectively obtain fine-grained interaction information between drugs and targets. Through experiments on three well-known benchmark datasets, we demonstrate that our proposed FusionDTI model outperforms eight state-of-the-art baselines in DTI prediction, particularly in the more realistic cross-domain scenario. Additionally, we show that the attention weights of the token-level fusion module in our model can highlight potential binding sites, providing a certain level of explainability.

**Limitations:** Even if our proposed model identifies potentially useful DTI, these predictions need to be validated by wet experiments, a time-consuming and expensive process.

**Potential impacts:** We have shown that FusionDTI is effective and efficient in screening for possible DTI in large-scale data as well as in locating potential binding sites in the process of drug design. However, it is not directly applicable to human medical therapy and other biomedical interactions because it lacks clinical validation and regulatory approval for medical use.

For future studies, we aim to investigate token-level interaction in more detail and to apply it to other biomedical scenarios, such as drug-drug interactions and protein-protein interactions.

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

# A APPENDIX

## A.1 HYPERPARAMETER OF FUSIONDTI

FusionDTI is implemented in Python 3.8 and the PyTorch framework (1.12.1)[1]. The computing device we use is the NVIDIA GeForce RTX 3090. In the "Experimental Setup and Results" section, we only present experiment results based on the BindingDB dataset, as the performance trends are identical to the BioSNAP dataset and the Human dataset. Table 6 shows the parameters of the FusionDTI model and Table 7 lists the notations used in this paper with descriptions.

Table 6: Configuration Parameters

| Module | Hyperparameter | Value |
|---|---|---|
| Mini-batch | Batch size | 64 (options: 64, 128) |
| Drug Encoder | PLM | HUBioDataLab/SELFormer |
| Protein Encoder | PLM | westlake-repl/SaProt_650M_AF2 |
| BAN | Heads of bilinear attention | 3 |
| | Bilinear embedding size | 512 (options: 32, 64, 128, 256, 512, 768) |
| | Sum pooling window size | 2 |
| CAN | Attention heads | 8 |
| | Hidden dimension | 512 (options: 32, 64, 128, 256, 512, 768) |
| | Integration strategies | Mean pooling (options: Mean pooling, CLS) |
| | Group size | 1 (options: from 1 to 512) |
| MLP | Hidden layer sizes | (1024, 512, 256) |
| | Activation | Relu (options: Tanh, Relu) |
| | Solver | AdamW |
| | | (options: AdamW, Adam, RMSprop, Adadelta, LBFGS) |
| | Learning rate scheduler | CosineAnnealingLR |
| | | (options: CosineAnnealingLR, StepLR, ExponentialLR) |
| | Initial learning rate | 1e-4 (options: from 1e-3 to 1e-6) |
| | Maximum epoch | 200 |

## A.2 DATASET SOURCES

All the data used in this paper are from public sources. The statistics of the experimental datasets are presented in Table 8.

1. The BindingDB (Gilson et al., 2016) dataset is a web-accessible database of experimentally validated binding affinities, focusing primarily on the interactions of small drug-like molecules and proteins. The BindingDB source is found at `https://www.bindingdb.org/bind/index.jsp`.

2. The BioSNAP (Zitnik et al., 2018) dataset is created from the DrugBank database (Wishart et al., 2008). It is a balanced dataset with validated positive interactions and an equal number of negative samples randomly obtained from unseen pairs. The BioSNAP source is found at `https://github.com/kexinhuang12345/MolTrans`.

---

[1] `https://pytorch.org/`

Table 7: Notations and Descriptions

| Notations | Description |
|---|---|
| $\mathbf{D}$ | Drug feature |
| $\mathbf{P}$ | Target feature |
| $\mathbf{q} \in \mathbb{R}^K$ | weight vector for bilinear transformation |
| $Att \in \mathbb{R}^{\rho \times \phi}$ | Bilinear attention maps in BAN |
| $\mathbf{U} \in \mathbb{R}^{N \times K}$ | Transformation matrix for drug features |
| $\mathbf{V} \in \mathbb{R}^{M \times K}$ | Transformation matrix for target features |
| $\mathbf{g}$ | The number of tokens per group |
| $\mathbf{D}^* \in \mathbb{R}^{m \times h}$ | Fused drug representations in token-level interaction |
| $\mathbf{P}^* \in \mathbb{R}^{n \times h}$ | Fused target representations in token-level interaction |
| $\mathbf{Q}_d, \mathbf{K}_d, \mathbf{V}_d \in \mathbb{R}^{m \times h}$ | Queries, keys, and values for the drug in token-level interaction |
| $\mathbf{Q}_p, \mathbf{K}_p, \mathbf{V}_p \in \mathbb{R}^{n \times h}$ | Queries, keys, and values for target in token-level interaction |
| $\mathbf{W}_q^d, \mathbf{W}_k^d, \mathbf{W}_v^d \in \mathbb{R}^{H \times h}$ | Projection matrices for drug queries, keys, and values |
| $\mathbf{W}_q^p, \mathbf{W}_k^p, \mathbf{W}_v^p \in \mathbb{R}^{h \times h}$ | Projection matrices for target queries, keys, and values |
| $\mathbf{F}$ | drug-target joint representation |
| $p \in [0, 1]$ | output interaction probability |
| $H$ | Number of attention heads in token-level interaction |
| $m, n$ | Sequence lengths for drug and protein respectively |
| $h$ | Hidden dimension in token-level interaction |

3. The Human (Liu et al., 2015; Chen et al., 2020) dataset includes highly credible negative samples. The balanced version of the Human dataset contains the same number of positive and negative samples. The Human source is found at `https://github.com/lifanchen-simm/transformerCPI`.

Table 8: Dataset Statistics

| Dataset | # Drugs | # Proteins | # Interactions |
|---|---|---|---|
| BindingDB | 14,643 | 2,623 | 49,199 |
| BioSNAP | 4,510 | 2,181 | 27,464 |
| Human | 2,726 | 2,001 | 6,728 |

### A.3 How to Obtain the Structure-aware (SA) Sequence of a Protein and the SELFIES of a Drug?

To obtain the SA sequence of a protein, the first step is to obtain Uniprot IDs from the UniProt website using information such as the amino acid sequences or protein names, and then save these IDs in a comma-delimited text file. Subsequently, we use the UniProt IDs to fetch the relevant 3D structure file (.cif) from AlphafoldDB (Varadi et al., 2022) using Foldseek. The SA vocabulary of the protein can then be generated from this 3D structure file.

For drugs, the SELFIES could be derived from SMILES strings. This conversion requires specific Python packages, and upon installation, the SELFIES strings can be generated through appropriate scripts. For more detailed procedures, including the necessary code, please refer to our submission file.

Notably, our submission of supplementary material contains step-by-step descriptions and code for generating the SA sequences and SELFIES.

### A.4 Baselines

We compare the performance of FusionDTI with the following eight models on the DTI task.

1. Support Vector Machine (Cortes & Vapnik, 1995) on the concatenated fingerprint ECFP4 (Rogers & Hahn, 2010) (extended connectivity fingerprint, up to four bonds) and PSC (Cao et al., 2013) (pseudo-amino acid composition) features.

2. Random Forest (Ho, 1995) on the concatenated fingerprint ECFP4 and PSC features.

3. DeepConv-DTI (Lee et al., 2019) uses a fully connected neural network to encode the ECFP4 drug fingerprint and a CNN along with a global max-pooling layer to extract features from the protein sequences. Then the drug and protein features are concatenated and fed into a fully connected neural network for the final prediction.

4. GraphDTA (Nguyen et al., 2021) uses GNN for the encoding of drug molecular graphs, and a CNN is used for the encoding of the protein sequences. The derived vectors of the drug and protein representations are directly concatenated for interaction prediction.

5. MolTrans (Huang et al., 2021) uses a transformer architecture to encode the drugs and proteins. Then a CNN-based fusion module is adapted to capture DTI interactions.

6. DrugBAN (Bai et al., 2023) use a Graph Convolution Network and 1D CNN to encode the drug and protein sequences. Then a bilinear attention network (Kim et al., 2018) is adopted to learn pairwise interactions between the drug and protein. The resulting joint representation is decoded by a fully connected neural network.

7. BioT5 (Pei et al., 2023) is a cross-modeling model in biology with chemical knowledge and natural language associations.

8. SiamDTI (Zhang et al., 2024) is a double-channel network structure to acquire local and global protein information for cross-field supervised learning.

## A.5 CASE STUDY

The top three predictions (PDB ID: 6QL2 Kazokaitė et al. (2019), 5W8L Rai et al. (2017) and 4N6H Fenalti et al. (2014)) of the co-crystalized ligands are derived from Protein Data Bank (PDB) Berman et al. (2007). Following the setup of the DrugBAN case study, we only choose X-ray structures with a resolution greater than 2.5 Å corresponding to human proteins. In addition, the co-crystalized ligands are required to have $pIC_{50} \leq 100$ nM and are not part of the training dataset. As shown in Figure 9, we summarise all drug-target interactions predicted by the DrugBAN and FusionDTI for the three sample pairs in the case study.

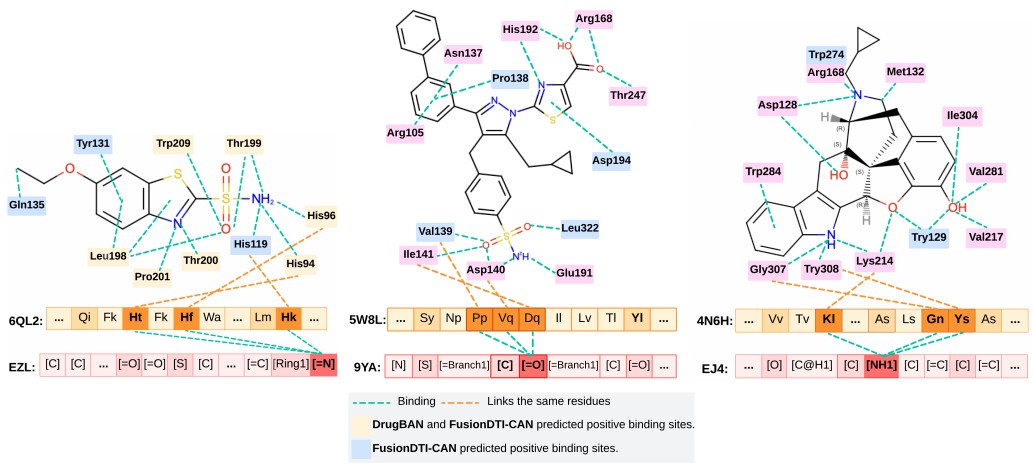

Figure 9: **FusionDTI predictions**: EZL - 6QL2, 9YA - 5W8L and EJ4 - 4N6H

## A.6 TIME COMPLEXITY ANALYSIS

The feature dimensions of the representations generated by different PLM encoders are fixed, but the size of the feature dimensions may not be the same. Therefore, in order to fuse protein and drug

Table 9: Time complexity and parameters comparison of BAN and CAN.

| Fusion module | Complexity (O) | Parameters |
|---|---|---|
| BAN | $O(\rho \cdot \phi \cdot K)$ | 790k |
| CAN | $O(m \cdot n \cdot h)$ | 1572k |

representations, we use two linear layers to keep the representations' feature dimension equal to the token length (512).

The time complexity of BAN depends on the computation of bilinear interaction maps. The bilinear attention involves a Hadamard product and further matrix operations as given in Equation (2). The computation of $U^T P$ and $V^T D$ requires $O(N \cdot \rho \cdot K)$ and $O(M \cdot \phi \cdot K)$ operations, respectively. Here, $K$ denotes the dimensionality of the transformation, which is the rank of the feature space to which the protein and drug features are projected. When the token length is equal to the feature dimension and the dimensions of transformation are two times either, the overall time complexity is $O(\rho \cdot \phi \cdot K)$.

For the token-level interaction in the DTI task, the time complexity is also markedly influenced by the attention mechanisms. It also satisfies the condition that the token length is equal to the feature dimension of the drug and protein. With multi-head attention heads ($H = 8$), the complexity for computing the queries, keys, and values in the Equation (6) and (7), as well as the softmax attention weights, is given by $O(H \cdot n \cdot m \cdot h)$, where $m$ and $n$ represents the token lengths for the drug and protein, respectively, and $h$ is the hidden dimension. Since each head contributes its own set of computations and the attention mechanism operates over all tokens, the $m \cdot n$ term (stemming from the softmax operation across the token length) becomes significant. This leads to a total time complexity of $O(m \cdot n \cdot h)$ per batch for the attention mechanism.

From the above analysis of the time complexity of the two fusion strategies, the time complexity of CAN is lower than BAN in the case of the same input protein and drug features. BAN is markedly affected by the transformation dimension $K$. When the $K$ is larger than the token and feature dimension, the time complexity of BAN is higher than CAN. However, we observe that the number of parameters in BAN is smaller than that of CAN via the Pytroch package, as shown in Table 9.

