# OpenReview forum: "FusionDTI: Fine-grained Binding Discovery with Token-level Fusion for Drug-Target Interaction"
_ICLR.cc/2025/Conference — ICLR 2025 Conference Withdrawn Submission_

### Official Review · Reviewer_2qY5 · 2024-10-31

**Soundness:** 2
**Presentation:** 3
**Contribution:** 2
**Rating:** 3
**Confidence:** 5

**Summary:**

The paper presents FusionDTI, a deep learning architecture for drug-target interaction (DTI) prediction that aims to capture fine-grained binding patterns between drug atoms and protein residues. The model's architecture integrates two specialized pre-trained language models: SELFormer for drug molecule encoding (using SELFIES representation) and Saport for protein sequence processing (using structure-aware vocabulary). The core contribution lies in the token-level fusion module, implemented through two variants: Bilinear Attention Network (BAN) and Cross Attention Network (CAN), designed to model detailed interaction patterns between molecular components.

The authors evaluate the model on three established DTI benchmark datasets using both in-domain and cross-domain validation protocols. Comparative analysis against eight baseline methods demonstrates competitive performance, with the CAN fusion module showing superior capability in capturing fine-grained interactions compared to BAN. The authors provide interpretability analysis through case studies that align with known binding site information from crystallographic structures.

While the implementation is technically sound and shows incremental improvements over existing methods, the primary innovation lies in the integration of established techniques rather than fundamental methodological advances in DTI prediction.

**Strengths:**

Well-engineered integration of state-of-the-art components

**Weaknesses:**

1. **Architectural Considerations**
        - The fusion module's novelty could be better justified beyond combining existing approaches
        - There is a contradiction in the description of model flexibility: it claims that the encoder can be replaced but relies on specific SELFIES and SA representations
2. **Methodological Aspects**
        - The dataset selection and splitting strategy, while valid, follows previous work (DrugBAN) without significant adaptation
        - The evaluation metrics suite could be expanded to include F1-score and Matthews Correlation Coefficient
3. **Experimental Validation**
        - Case studies could be more innovative and differentiated from DrugBAN
        - The same evaluation metrics in DrugBAN should be shown

**Questions:**

1. **Methodology**
        - What motivated the selection of unpublished work (SiamDTI) as a baseline?
        - How does protein sequence length impact prediction accuracy?
        - Please specify the dataset context for results in Figures 5-7
2. **Theoretical Foundation**
        - What evidence supports the correlation between token-level interactions and actual molecular binding sites?
        - How was the choice between BAN and CAN architectures motivated?
3. **Practical Applications**
        - Has the model been validated in real-world drug discovery scenarios like virtual screening?
        - How can this approach be extended to other types of molecular interactions beyond DTI?

---

> ### Author Response · Authors · 2024-11-20
> **Response to the Review Comments.**
>
> ## Weaknesses:
>
> **Weakness 1**: Architectural Considerations - The fusion module's novelty could be better justified beyond combining existing approaches - There is a contradiction in the description of model flexibility: it claims that the encoder can be replaced but relies on specific SELFIES and SA representations.
>
> **Reply**: Thank you for your comments. However, we would argue that the work done in this paper is novel. We sincerely hope that our **general responses** have addressed your concerns.
>
> **Fusion module's novelty**:
>
> We believe the fusion module's novelty is well-justified, as it goes beyond merely combining existing approaches by incorporating biomedical principles to address fine-grained limitations in existing models. Specifically, we integrate Fine-grained Representation and Innovation Strategy, ensuring token-level fusion to capture fine-grained interactions between atoms and amino acids. Moreover, we incorporate biological knowledge to perform Granular Interaction Validation, systematically comparing token, substructure, and molecular-level interactions. This approach demonstrates the effectiveness of token-level fusion in improving prediction accuracy, as evidenced in Figure 5.
>
> **Model Flexibility**:
>
> Our proposed model is flexible in that the encoder can be replaced with any pre-trained model capable of generating token-level representations, such as the recent SELFIES-BART (16 Oct 2024). While the current implementation leverages SELFIES and SA representations, other representations could also be explored, provided they capture fine-grained interactions effectively. For instance, alternatives like SMILES-BERT for drugs or amino acid sequence-based models for proteins could serve as replacements in scenarios where SELFIES or SA are unavailable. We will clarify this point more clearly in the revised paper.
>
> **Weakness 2**: Methodological Aspects - The dataset selection and splitting strategy, while valid, follows previous work (DrugBAN) without significant adaptation - The evaluation metrics suite could be expanded to include F1-score and Matthews Correlation Coefficient.
>
> **Reply**:
> 1. Most previous DTI tasks used in-domain splitting strategies, which often lack practical relevance. Our study goes well beyond that, adding for example a cross-domain setting. Note that cross-domain data segmentation strategies have only been used in the DTI literature for a short period of time, and adapting them, as we did in this paper, to realistic biomedical scenarios is still a great challenge.
>
> 2. Thank you for your suggestion of evaluation metrics. The following is the latest experiment results with F1-score and Matthews Correlation Coefficient as evaluation metrics. We will add these new results in the revised paper:
>
> **In-domain Performance**
>
> | Dataset | Model  | F1-score | MCC |
> |---------|---------|---------|---------|
> | BindingDB  | DrugBAN | 0.901±0.004     | 0.872±0.005 |
> |            | FusionDTI-BAN    | 0.934±0.002     | 0.900±0.003 |
> |            | FusionDTI-CAN    | **0.963±0.012** | **0.925±0.023**  |
> | BioSNAP    | DrugBAN | 0.830±0.009     | 0.719±0.007 |
> |            | FusionDTI-BAN  | 0.857±0.001     | 0.724±0.001 |
> |            | FusionDTI-CAN  | **0.890±0.002** | **0.778±0.002**  |
> | Human      | DrugBAN  | 0.903±0.003     | 0.810±0.004 |
> |            | FusionDTI-BAN    | 0.934±0.002     | 0.870±0.003 |
> |            | FusionDTI-CAN    | **0.948±0.002** | **0.905±0.045**  |
>
> ---
>
> **Out-domain Performance**
>
> | Dataset | Model   | F1-score  | MCC  |
> |----------|-----------|----------|----------|
> | BindingDB  | DrugBAN  | 0.582±0.030 | 0.187±0.031|
> |            | FusionDTI-BAN  | 0.587±0.002 | 0.276±0.003 |
> |            | FusionDTI-CAN  | **0.601±0.005** | **0.302±0.005**|
> | BioSNAP    | DrugBAN  | 0.587±0.005  | 0.219±0.017|
> |            | FusionDTI-BAN  | 0.597±0.001 | 0.254±0.010|
> |            | FusionDTI-CAN | **0.602±0.012** | **0.268±0.011**|
> | Human  | DrugBAN       | 0.711±0.030| **0.261±0.010**|
> |            | FusionDTI-BAN | 0.725±0.002     | 0.212±0.011|
> |            | FusionDTI-CAN | **0.736±0.010** | 0.238±0.013 |
>
> **Weakness 3**: Experimental Validation - Case studies could be more innovative and differentiated from DrugBAN - The same evaluation metrics in DrugBAN should be shown.
>
> **Reply**: Our proposed model allows us to directly visualise binding sites using attention maps without the aid of docking visualisation tools. The case study examines three drug-target pairs with ground truth from Protein Data Bank (PDB) for easy comparison with DrugBAN. Notably, all three pairs included binding sites and are validated by wet experiments. Therefore, we do actually compare the predicted binding sites with the ground truth, which shows that our model can predict more binding sites than DrugBAN. To further address the reviewer’s suggestion,  we will also provide an additional ten pairs of comparisons between predicted binding sites and real data in the revised version.

---

> > ### Author Response · Authors · 2024-11-20
> > **Response to the Review Comments.**
> >
> > ## Questions:
> >
> > **Question 1**: Methodology - What motivated the selection of unpublished work (SiamDTI) as a baseline? - How does protein sequence length impact prediction accuracy? - Please specify the dataset context for results in Figures 5-7
> >
> > **Reply**:
> >
> > 1. SiamDTI is one of the latest DTI prediction models that outperforms DrugBAN and is available as open source code. We reproduced the experiments and verified the results. Hence, since it is available, we were of the opinion that it is necessary to use this cutting-edge model as a baseline.
> > 2. The pre-trained language models have a fixed token length of 512. For protein sequences longer than 512, some information is inevitably lost, which can impact accuracy. However, this is not the primary focus of our work.
> > 3. The dataset used for Figures 5-7 is the BindingDB dataset.
> > The revised paper will include the details above.
> >
> > **Question 2**: Theoretical Foundation - What evidence supports the correlation between token-level interactions and actual molecular binding sites? - How was the choice between BAN and CAN architectures motivated?
> >
> > **Reply**:
> > 1. In the abstract and introduction sections, we highlighted the binding principle of DTI in biomedicine. Specifically, DTI refers to the binding of specific drug atoms to key amino acids of a protein rather than substructures. In the Protein Data Bank (PDB) dataset, the 3D View of DTI corresponds to specific atoms and individual amino acids.
> >
> > 2. Below, we provide a detailed explanation of how the choice between the BAN and CAN architectures was motivated, supported by results in our experiments:
> >
> > **Effectiveness Comparison (Sections 4.2 and 4.3)**:
> > As shown in Tables 1 and 2, FusionDTI-CAN consistently achieves a superior performance on both in-domain and cross-domain datasets, with higher AUROC and AUPRC values compared to FusionDTI-BAN.
> > In Figure 3, FusionDTI-CAN outperforms BAN as the feature dimensions increase, maintaining its performance advantage due to its ability to capture more nuanced token-level interactions.
> >
> > **Efficiency Comparison (Section 4.4)**:
> > Figure 4 highlights that FusionDTI-BAN is the most efficient model in terms of time consumption due to its simpler bilinear interaction mechanism. This makes BAN a preferable choice for scenarios requiring high-speed predictions.
> >
> > **Fine-grained Interaction Modelling (Section 4.6)**:
> > In Figure 5, we show that CAN captures more detailed interactions by consistently achieving better accuracy across various fusion scales. CAN’s ability to integrate dependencies within and across token representations allows it to better model fine-grained drug-target interactions compared to BAN.
> >
> > Based on these results, CAN is recommended for scenarios where the predictive accuracy and explainability are the priorities, while BAN is better suited for applications requiring faster computation. We will make this clearer in the revised paper.
> >
> > **Question 3**: Practical Applications - Has the model been validated in real-world drug discovery scenarios like virtual screening? - How can this approach be extended to other types of molecular interactions beyond DTI?
> >
> > **Reply**:
> > 1.  While our proposed model has not been applied in a real-world scenario (such real deployments are not easy in our country where there is a lengthy clinical and legal framework to go through), please note that we do simulate a real-world screening scenario by splitting the dataset such that the training and test data contain distinct drugs and targets. This setup prevents using known drug or target features during predictions on the test data, closely resembling virtual screening conditions.
> > 2. This approach can be extended to other types of molecular interactions beyond DTI, such as drug-drug interactions (DDI) and protein-protein interactions (PPI). By adapting the input data and modifying the learning objectives, the proposed model can be retrained to predict different interaction types, potentially providing insights into combination therapies or understanding protein interaction networks. This is the direction of our future work.

---

> > ### Comment · Reviewer_2qY5 · 2024-11-25
> >
> > Thank you for your response. While your clarifications and additional experimental results partially address some concerns, several critical aspects still require attention.
> >
> > Regarding Weakness 1 (Architectural Considerations), the visualization tools and integration of existing components demonstrate good engineering, but the fundamental innovation level expected for ICLR remains inadequately demonstrated. The claimed advantages in protein structure consideration appear to be primarily inherited from the underlying language models rather than from novel architectural contributions. The model's flexibility claim requires further substantiation, particularly concerning the interdependence between the fusion mechanism and the specific representations used.
> >
> > Concerning Weakness 2 (Methodological Aspects), I appreciate the newly provided F1-score and MCC results, which demonstrate promising performance. However, your characterization of cross-domain splitting as a recent innovation in DTI is inaccurate. This approach was established by the authors of DrugBAN through their hierarchical clustering split methodology [1] in 2021, and has since become a standard practice in the field. The experimental setup needs to explore more challenging and realistic splitting scenarios, potentially incorporating newer datasets that reflect current biological challenges.
> >
> > For Weakness 3 (Experimental Validation), I look forward to reviewing the additional ten pairs of binding site comparisons promised in the revision. Please ensure these new cases demonstrate diverse binding mechanisms and challenging scenarios that differentiate your approach from existing methods.
> >
> > Regarding the Methodology Questions, several aspects require deeper examination. First, while you explained the rationale for including SiamDTI, I still recommend comparison with published 2024 baselines. Second, the protein sequence length limitation warrants quantitative analysis, including the distribution of sequence lengths in real-world DTI data and performance analysis stratified by sequence length. A clear discussion of how information loss in longer sequences affects prediction accuracy is essential for understanding the method's practical limitations.
> >
> > Concerning the Theoretical Foundation Questions, the token-level interaction claims need stronger theoretical foundation supported by biochemical literature, beyond empirical observations. The choice between BAN and CAN architectures would benefit from theoretical justification beyond performance differences. Your response primarily focused on experimental results rather than providing the requested theoretical underpinning.
> >
> > Finally, regarding the Practical Applications Questions, your interpretation of virtual screening appears to conflate it with clinical trials. Virtual screening is a computational methodology for preliminary drug discovery that doesn't require clinical or legal frameworks. The paper would benefit from demonstrating how FusionDTI could be applied in standard virtual screening protocols, such as large-scale compound library screening and hit identification workflows.
> >
> > References:
> > [1] Bai, P. et al. (2021) Hierarchical clustering split for low-bias evaluation of drug-target interaction prediction. In 2021 IEEE International Conference on Bioinformatics and Biomedicine (BIBM) (pp. 641-644). IEEE.

---

### Official Review · Reviewer_NQpv · 2024-11-01

**Soundness:** 3
**Presentation:** 3
**Contribution:** 2
**Rating:** 5
**Confidence:** 5

**Summary:**

The authors present FusionDTI, a drug-protein interaction prediction model developed to enhance fine-grained interaction learning. The model introduces a token-level (atoms for drugs, residues for proteins) fusion module based on bilinear attention (BAN) or cross-attention (CAN) mechanisms. It leverages pretrained encoders, Saprot for ligands and SELFormer for proteins, to capture comprehensive molecular features. Experimental results on three benchmark datasets demonstrate robust improvements over competitive baselines. The authors also include extensive ablation studies to validate the uniqueness and effectiveness of each component in FusionDTI. Furthermore, a case study illustrates how the fine-grained interaction learning enhances model interpretability.

**Strengths:**

- The fine-grained interaction learning is the performance bottleneck of DTI prediction models, which is valuable for designing new strategies.
- The fusion module is clearly defined.
- A comprehensive ablation study is conducted to examine each part of the model, including different pretrained models, fusion modules, and hyperparameters.

**Weaknesses:**

- While the study is technically sound, many of the components used in FusionDTI, such as the cross attention and bilinear attention mechanisms, have been well studied in previous DTI research as acknowledged and cited by the authors. FusionDTI appears to be more of **an integration of known pretrained encoders and existing interaction modules**. We may not gain new insights from this study into improving the computational simulation of drug-protein binding, as using attention mechanisms for atom-residue interactions is already a widely adopted strategy. This raises questions about the study’s methodological novelty.

- The model interpretation aspect of the study has several limitations:

    - The selected cases for demonstrating FusionDTI's interpretability are **not representative**. Readers would be interested not only in the very top predictions but also in moderate and poor predictions (like some bad case analysis) because there is no clear threshold or metric provided to assess whether a prediction is good enough for practical interpretative use.

     - Some inconsistencies in the case study results need to be addressed. For instance, GLN92 is highlighted in Table 5 but does not appear in Figure 9. Please double check that.

    - Incorporating a **binding structure visualization analysis** would greatly enhance the comparison between the predicted interactions and the experimentally validated interactions. It would be also helpful for determining which one (FusionDTI or DrugBAN) aligns best with the known interactions.

    - A better solution could involve **quantifying the attention visualization results**. For example, calculating how much of key residues or interactions are highlighted by attention weights on a larger scale dataset, such as PoseBusters or CASF, would help to verify the tool’s effectiveness in elucidating drug-protein binding modes.

**Questions:**

- What specific **selection criteria or threshold for attention weights** were used to determine the predicted interactions between ligand atoms and protein residues?
- The accuracy results for the Human dataset are missing.

---

> ### Author Response · Authors · 2024-11-20
> **Response to the Review Comments.**
>
> ## Weaknesses:
>
> **Weaknesses 1**:
>
> **Reply**: Thank you for your comments. However, we would argue that the work done in this paper is novel. We sincerely hope that our **general responses** have addressed your concerns.
>
> **Weaknesses 2**:
>
> **Weaknesses 2.1**: The selected cases for demonstrating FusionDTI's interpretability are not representative. Readers would be interested not only in the very top predictions but also in moderate and poor predictions (like some bad case analysis) because there is no clear threshold or metric provided to assess whether a prediction is good enough for practical interpretative use.
>
> **Reply**: Thank you for your suggestion. We examine the three drug-target pairs with ground truth from Protein Data Bank (PDB) to allow for comparison with DrugBAN. Some binding sites without evidence of support (poor predictions) are also highlighted by attention mapping. In the submitted code, we include a notebook file that enables users to visualise specific attention maps for both strong and weak predictions. Since there is no well-established threshold for determining prediction quality, we adopt a ranking-based approach. This allows the performance of the model to be explored, providing a way of assessing interpretability in the top, medium and poor cases.
>
> **Weaknesses 2.2**: Some inconsistencies in the case study results need to be addressed. For instance, GLN92 is highlighted in Table 5 but does not appear in Figure 9. Please double check that.
>
> **Reply**: Thanks for your comments. The highlighted amino acid (GLN92) validation is based on a published paper but is not present in the PDB database. In Figure 9, we only labelled the results predicted by the model that are validated by the PDB database. We will better explain the issue in Figure 9’s caption in the revised version.
>
> **Weaknesses 2.3**: Incorporating a binding structure visualization analysis would greatly enhance the comparison between the predicted interactions and the experimentally validated interactions. It would be also helpful for determining which one (FusionDTI or DrugBAN) aligns best with the known interactions.
>
> **Reply**: Our proposed model allows for direct visualisation of binding sites using attention maps without the aid of docking visualisation tools. Therefore, we compare the predicted binding sites with the ground truth and show that our model can predict more binding sites than DrugBAN (c.f. Table 5). Moreover, please note that the goal of our task is to identify whether a drug and a target will interact rather than predict the exact binding state of their docking.
>
> **Weaknesses 2.4**: A better solution could involve quantifying the attention visualization results. For example, calculating how much of key residues or interactions are highlighted by attention weights on a larger scale dataset, such as PoseBusters or CASF, would help to verify the tool’s effectiveness in elucidating drug-protein binding modes.
>
> **Reply**: Thank you for your suggestion. Quantifying attention visualisation results is extremely time-consuming since processing known binding sites requires manual manipulation of the data, so we will not be able to provide more visualisation results during the author-response process. However, to address the reviewer’s suggestion, we will be providing at least ten pairs of comparisons between predicted binding sites and real data in the final version.
>
> ## Questions:
>
> **Question 1**: What specific selection criteria or threshold for attention weights were used to determine the predicted interactions between ligand atoms and protein residues?
>
> **Reply**: There is currently no well-known threshold for measuring interaction strength, so we identified binding sites of specific atoms and amino acids in the attention map by ranking. We will clarify the issue in the revised paper.
>
> **Question 2**: The accuracy results for the Human dataset are missing.
>
> **Reply**: Thank you for your suggestion. The absence of accuracy results for the Human dataset aligns with the standard practice in existing studies, including advanced models like DrugBAN and BioT5, which focus on AUC and AUPRC as primary metrics due to their ability to provide a more comprehensive evaluation of imbalanced datasets like DTI. These metrics are particularly relevant when AUC and AUPRC values approach 99%, making accuracy less informative for distinguishing performance. However, we appreciate the reviewer’s suggestion and the importance of a comprehensive evaluation. Hence, we will include the accuracy results for the Human dataset in the revised version. The results are also shown below.
>
> **In-domain Performance (Human)**
>
> | Model | Accuracy|
> |--------|---------|
> | DrugBAN | 0.930±0.004|
> | FusionDTI-BAN | 0.938±0.003|
> | FusionDTI-CAN | 0.947±0.002|
> ---
> **Out-domain Performance (Human)**
>
> | Model | Accuracy|
> |---------|------------|
> | DrugBAN | 0.709±0.005|
> | FusionDTI-BAN | 0.731±0.003|
> | FusionDTI-CAN |0.738±0.002|

---

### Official Review · Reviewer_ocCD · 2024-11-02

**Soundness:** 2
**Presentation:** 3
**Contribution:** 3
**Rating:** 5
**Confidence:** 4

**Summary:**

This paper presents FusionDTI, a model for drug-target interaction (DTI) prediction that claims to improve interpretability through fine-grained interactions between drug components and protein residues. The authors leverage the two existing backbone models (BAN and CAN) to achieve the token-level interaction and finally search for the bind site with these tokens.

**Strengths:**

- The paper is well-written and easy to follow.
- The proposed token-fusion (TF) strategy is straightforward yet reasonable.
- The experimental results and case studies demonstrate excellent prediction performance and strong interpretability.

**Weaknesses:**

See Questions.

**Questions:**

- The authors leverage two existing backbone models (BAN and CAN) to achieve token-level interactions and ultimately search for binding sites using a dense linkage of all these tokens, which appears to be both simple and computationally intensive. Notably, DrugBAN has already employed BAN for a quite similar fusion objective, with the only difference being that the basic element is the substructure. Therefore, the novelty proposed by the authors is concerning.
- The paper lacks a theoretical contribution regarding the proposed method for the DTI task.
- In the case study for searching for binding sites, FusionDTI-CAN is adopted for comparison with DrugBAN. It seems more reasonable to use FusionDTI-BAN for a fair comparison, which raises confusion. So why not choose BAN as backbone model?
- Although the TF module is useful, its computational complexity clearly indicates that it is quite time-consuming. What will happen if the model is faced with larger drug molecules or larger protein sequence datasets?
- It should be clear whether the improvements benefit from the pre-trained language models.  The ablation results of w/o LLM pre-trained feature is needed.

---

> ### Author Response · Authors · 2024-11-20
> **Response to the Review Comments.**
>
> **Questions 1**: The authors leverage two existing backbone models (BAN and CAN) to achieve token-level interactions and ultimately search for binding sites using a dense linkage of all these tokens, which appears to be both simple and computationally intensive. Notably, DrugBAN has already employed BAN for a quite similar fusion objective, with the only difference being that the basic element is the substructure. Therefore, the novelty proposed by the authors is concerning.
>
> **Reply**: Thank you for your comments. However, we would argue that the work done in this paper is novel. We sincerely hope that our **general responses** have addressed your concerns.
>
> **Questions 2**: The paper lacks a theoretical contribution regarding the proposed method for the DTI task.
>
> **Reply**: As we explain in our general response, the existing models in the literature are not sufficiently fine-grained. Despite remarkable advances in recent DTI models, significant challenges remain, particularly in aligning the model design with biomedical principles (token-level interaction). Our proposed solutions can be characterised as conceptual contributions advancing the existing literature through an innovative new framework for the DTI task. Below we list the **research questions** and how they can be addressed to advance DTI discovery.
>
> **1. How can fine-grained representations improve DTI predictions?**
>
> Existing models rely on SMILES and amino acid sequences, which lack atomic-level precision. We address this by using SELFIES and structure-aware protein sequences ​(FusionDTI).
>
> **2. What is the best way to capture sufficiently fine-grained interactions?**
>
> We propose a token-level fusion with pre-trained encoders, which captures token-level interactions overlooked by the existing models in the literature​.
>
> **3. Can token-level fusion improve both accuracy and explainability?**
>
> Our method demonstrates a superior predictive performance on both in-domain and cross-domain datasets and facilitates the highlighting of binding sites, as shown and validated through case studies​.
>
> Responses to these address critical challenges in DTI modelling, offering a novel, explainable framework for fine-grained interaction prediction and advancing the field.
>
> **Questions 3**: In the case study for searching for binding sites, FusionDTI-CAN is adopted for comparison with DrugBAN. It seems more reasonable to use FusionDTI-BAN for a fair comparison, which raises confusion. So why not choose BAN as backbone model?
>
> **Reply**: Thank you for your suggestion. In the case study, we preferred to show whether our proposed model can predict more binding sites. However, we will also compare FusionDTI-BAN with DrugBAN for completeness and put the results in the appendix section of the revised version.
>
> **Questions 4**: Although the TF module is useful, its computational complexity clearly indicates that it is quite time-consuming. What will happen if the model is faced with larger drug molecules or larger protein sequence datasets?
>
> **Reply**: For larger drug molecules or larger protein sequence datasets, the inference time will be the same for the TF module, since the output dimensions of the protein encoder and molecular encoder are fixed. We will add a clarification in the revised paper.
>
> **Questions 5**: It should be clear whether the improvements benefit from the pre-trained language models. The ablation results of w/o LLM pre-trained feature is needed.
>
> **Reply**: Thank you for your comment. While we did not conduct a specific ablation study without pre-trained features, the comparison between FusionDTI-BAN and DrugBAN indirectly demonstrates the benefits of the pre-trained encoders. FusionDTI-BAN, which leverages pre-trained features, consistently outperforms DrugBAN, which does not. To address the reviewer’s comment, we will also add the explicit ablation results in the appendix section of the revised version.

---

> > ### Comment · Reviewer_ocCD · 2024-11-25
> >
> > I appreciate the detailed response from the authors. However, my main concerns remain unaddressed:
> > * Novelty: FusionDTI appears simplistic and brute-force, lacking sufficient motivation. It just feels like an extension of the patch-based algorithm in ViT [1] to the pixel-based domain. Furthermore, a paper published in Nature Chemical Biology [2] argues that DTI is determined by the complex interactions between the important molecular substructures in the drug and binding sites in the protein sequence. This directly contradicts the author's statement in the global response, where it is stated that "the existing binding of DTI is drug single atom and individual amino acid residues," which undermines the motivation behind the approach.
> > * Theoretical Analysis: Theory does not equate to motivation, and the authors have merely reiterated their motivation and experimental results in the response. What I was hoping for was a more in-depth mathematical analysis of the value of the token-level interaction strategy.
> > * Case Study: This paper is closely related to DrugBAN, and in this context, if the authors intend to emphasize the superiority of their method over DrugBAN, a controlled-variable strategy is absolutely necessary. Moreover, since the authors have already completed the training of FusionDTI-BAN, visualizing the case results should be straightforward. Why not directly provide the comparison results?
> > * Computational Complexity: My concern lies in the computational resources required for TRAINGING on larger drug molecules or larger protein sequence datasets. The token-level interactions raise obvious concerns about increased computational burden.
> > * Ablation Studies: As the authors themselves mention, " FusionDTI-BAN, which leverages pre-trained features, consistently outperforms DrugBAN, which does not.”, this only intensifies my concern. The core innovation of this work is the token-level interaction strategy, so the performance improvement should not primarily rely on the pre-trained representations of LLMs. It is clear that this issue has not been adequately addressed by the authors.
> >
> > I feel that the authors have largely failed to directly address my concerns. As such, I will maintain my current rating.
> >
> > [1] Dosovitskiy, Alexey, et al. "An Image is Worth 16x16 Words: Transformers for Image Recognition at Scale." International Conference on Learning Representations. 2020. \
> > [2] Schenone, Monica, et al. "Target identification and mechanism of action in chemical biology and drug discovery." Nature chemical biology 9.4 (2013): 232-240.

---

### Official Review · Reviewer_qmCU · 2024-11-04

**Soundness:** 2
**Presentation:** 3
**Contribution:** 2
**Rating:** 3
**Confidence:** 4

**Summary:**

This paper presents FusionDTI, a new model designed to improve drug-target interaction (DTI) predictions. FusionDTI employs a token-level Fusion module to capture fine-grained interactions between drug atoms and protein amino acids. It utilizes the SELFIES representation for drugs and a structure-aware vocabulary for target proteins, while leveraging pre-trained language models to enhance understanding of complex relationships. Using the drug and protein embeddings, some existing embedding fusion strategies were evaluated.   Experiments demonstrate that FusionDTI outperforms eight state-of-the-art models, and its case study highlights potential binding sites, increasing the explainability of DTI predictions.

**Strengths:**

-Used large language models to extract both protein and drug features.
-Evaluated the performance of the model for both in-domains and out-of-domains.
-Explored potential interpretability of the model.

**Weaknesses:**

-Overall, the novelty of the approach is low. It is not novel to apply large language model to extract protein and drug features in DTI prediction. Many related work have been published.
-It is also not novel to use the applied fussion strategies for DTI prediction. Both of the fusion strategies have been widely used before.

**Questions:**

The case example appears overly simplistic. Is this prediction based on in-domain or out-of-domain data? I recommend conducting an analysis of out-of-domain cases. Additionally, the results would benefit from external validation. For instance, performing blind docking studies on the drug-protein pairs could confirm whether they truly interact as predicted, and visualizing the binding sites would provide further insights into the interaction.

---

> ### Author Response · Authors · 2024-11-20
> **Response to the Review Comments.**
>
> **Weaknesses**
>
> **Reply**: Thank you for your comments. However, we would argue that the work done in this paper is novel. We sincerely hope that our **general responses** have addressed your concerns.
>
> **Questions**:
>
> The case example appears overly simplistic. Is this prediction based on in-domain or out-of-domain data? I recommend conducting an analysis of out-of-domain cases. Additionally, the results would benefit from external validation. For instance, performing blind docking studies on the drug-protein pairs could confirm whether they truly interact as predicted, and visualizing the binding sites would provide further insights into the interaction.
>
> **Reply**: Thank you for your suggestion. To clarify, the three DTI pairs in our case study are based on out-of-domain (cross-domain) data, which means they belong to neither the training dataset nor the validation dataset. They are derived from the Protein Data Bank (PDB), which contains binding sites that wet experiments have validated. Therefore, we do actually compare the predicted binding sites with the ground truth data without additional validation. Note also that as Shown in Figure 8, our proposed model allows us to directly visualise binding sites using attention maps without the aid of docking visualisation tools. We will make this clearer in the revised paper.

---

> > ### Comment · Reviewer_qmCU · 2024-11-25
> >
> > I appreciate the authors' feedback. However, I still believe that 1) the novelty of the paper is relatively limited, although I am open to including relevant citations here; and 2) the case study presented is too simplistic. As such, I stand by my original score.

---

### Author Response · Authors · 2024-11-20
**General response to all reviewers.**

Thank you for your comments. However, we would argue that the work done in this paper is novel. Predicting drug-target interactions (DTIs) is a cornerstone of the drug discovery process, as it aids in identifying potential therapeutic targets and supports the development of novel drugs. In the following, we outline how our work extends beyond the existing literature.

## Fine-grained Challenges of Current DTI Models:

1. Existing DTI models predominantly use SMILES for drugs and amino acid sequences for proteins, which lack sufficient chemical and structural details critical for fine-grained interaction discovery.
2. The reliance on substructure representations (e.g., GNNs for drug SMILES, 3-mer sequences for proteins) fails to capture sufficiently fine-grained interactions, which is critical to capturing DTI. Specifically, the existing binding of DTI is drug single atom and individual amino acid residues, as demonstrated by structural data from the Protein Data Bank (PDB).

## Our Novel Contributions:

**Fine-grained Representation**: We utilise **SELFIES** for drugs and **Structure-Aware Sequences** for proteins, ensuring atomic-level precision and structural information during tokenization, addressing the existing limitations of SMILES and amino acid sequences.

**Innovation Strategy**: Our token-level fusion with pre-trained encoders enables the model to represent and integrate drug and protein sequences at a token level, focusing on interactions between individual atoms and amino acids—a gap not addressed by existing models such as *DrugBAN*.

**Granular Interaction Validation**: We are the first to systematically compare token, substructure, and molecular-level interactions through a cross-attention module, demonstrating that fine-grained fusion consistently enhances prediction accuracy as shown in Figure 5.

**Case Study is Explainable**: Through our case study, we predicted and validated three DTI pairs (not included in the training and validation datasets) in the Protein Data Bank, highlighting additional and new real binding sites compared to *DrugBAN*.

**Performance Highlights**: FusionDTI-CAN achieves a state-of-the-art performance on existing benchmark datasets based on both in-domain (e.g., BindingDB: accuracy of \( 0.961 \)) and cross-domain (e.g., BioSNAP: accuracy of \( 0.734 \)), significantly surpassing existing baselines.

---

> ### Author Response · Authors · 2024-11-22
> **This is a gentle reminder.**
>
> Dear reviewers,
>
> We sincerely hope that our responses have addressed your concerns and hope you will consider increasing your score. If we have left any notable points of concern overlooked, we would greatly appreciate your feedback, and we will attend to these points. Additionally, we will incorporate all the suggestions and discussions mentioned in the latest manuscript. Thanks again for your thoughtful review and consideration.

---

### Note · Authors · 2024-12-02

I have read and agree with the venue's withdrawal policy on behalf of myself and my co-authors.